# Multimodal Diagnostic Approaches to Advance Precision Medicine in Sarcopenia and Frailty

**DOI:** 10.3390/nu14071384

**Published:** 2022-03-26

**Authors:** David H. Lynch, Hillary B. Spangler, Jason R. Franz, Rebecca L. Krupenevich, Hoon Kim, Daniel Nissman, Janet Zhang, Yuan-Yuan Li, Susan Sumner, John A. Batsis

**Affiliations:** 1Division of Geriatric Medicine, Center for Aging and Health, University of North Carolina, Chapel Hill, NC 27599, USA; john.batsis@gmail.com; 2Division of Medicine and Pediatrics, University of North Carolina, Chapel Hill, NC 27599, USA; hillary.spangler@unchealth.unc.edu; 3Joint Department of Biomedical Engineering, University of North Carolina at Chapel Hill, North Carolina State University, Chapel Hill, NC 27599, USA; jrfranz@email.unc.edu (J.R.F.); rlkrup@email.unc.edu (R.L.K.); hoonkim@email.unc.edu (H.K.); 4Department of Radiology, University of North Carolina School of Medicine, Chapel Hill, NC 27599, USA; daniel_nissman@med.unc.edu (D.N.); janet.zhang@unchealth.unc.edu (J.Z.); 5Department of Nutrition, Gillings School of Global Public Health, University of North Carolina, Chapel Hill, NC 27599, USA; yuanyli4@unc.edu (Y.-Y.L.); susan_sumner@unc.edu (S.S.)

**Keywords:** sarcopenia, sarcopenic obesity, precision medicine, intramuscular fat

## Abstract

Sarcopenia, defined as the loss of muscle mass, strength, and function with aging, is a geriatric syndrome with important implications for patients and healthcare systems. Sarcopenia increases the risk of clinical decompensation when faced with physiological stressors and increases vulnerability, termed frailty. Sarcopenia develops due to inflammatory, hormonal, and myocellular changes in response to physiological and pathological aging, which promote progressive gains in fat mass and loss of lean mass and muscle strength. Progression of these pathophysiological changes can lead to sarcopenic obesity and physical frailty. These syndromes independently increase the risk of adverse patient outcomes including hospitalizations, long-term care placement, mortality, and decreased quality of life. This risk increases substantially when these syndromes co-exist. While there is evidence suggesting that the progression of sarcopenia, sarcopenic obesity, and frailty can be slowed or reversed, the adoption of broad-based screening or interventions has been slow to implement. Factors contributing to slow implementation include the lack of cost-effective, timely bedside diagnostics and interventions that target fundamental biological processes. This paper describes how clinical, radiographic, and biological data can be used to evaluate older adults with sarcopenia and sarcopenic obesity and to further the understanding of the mechanisms leading to declines in physical function and frailty.

## 1. Introduction: A Basic Model for Geroscience

Sarcopenia, defined as a loss of muscle mass, strength, and physical function [1,2], is a key contributor to the development of the frailty syndrome in older adults. Frailty is characterized by a loss of biological reserve and increased vulnerability to physiological stressors [3]. An aging demographic, and declining mortality rates have contributed to the rising rates of sarcopenia and frailty in the United States [4]. These changes have important implications for patients and the healthcare systems as they are associated with many adverse outcomes, including hospitalization, long-term care admission, mortality, and reduced quality of life [3,5,6].

The development of sarcopenia, obesity, and frailty results from multiple contributing factors, including biological changes with age, sedentary lifestyle, poor nutritional habits, multimorbidity, and acute illness (Figure 1) [1]. Additionally, unhealthy lifestyles have also led to rising rates of obesity and its associated adverse clinical outcomes, which has made it increasingly common for older adults to be living with a combination of these disorders. The pathophysiology of frailty in older adults is complex. Emerging evidence suggests that distinct phenotypes exist within the broader definition (discussed in detail in prior research and beyond the scope of this article [4,7]). This article will focus on physical frailty, defined as the presence of three of the five following components: weakness, slow gait speed, low physical activity, exhaustion, and weight loss. [8] Weakness is often the initial sign of developing physical frailty. Therefore, it is not surprising that biological factors that contribute to the development of physical frailty overlap significantly with those described for sarcopenia and obesity. [9] This shared pathway can be summarized as myocellular changes resulting in a dysregulation of metabolic, musculoskeletal, and hormonal pathways, resulting in progressive gains in fat mass and loss of lean mass and muscle strength, leading to the syndromes of sarcopenic obesity and physical frailty [7,10].

While the prevalence rates of sarcopenia, sarcopenic obesity, and frailty vary widely because of differing definitions [11], their increased recognition and clinical importance has led to a critical gap in translating research findings into practice. Sarcopenic obesity can be determined by defining *sarcopenia* using strength (measured by grip strength, chair stand) or muscle mass (lean body mass, appendicular lean mass) measured using body composition modalities (e.g., dual energy x-ray absorptiometry [DEXA], magnetic resonance imaging [MRI]), and obesity using body mass index, waist circumference, or body fat [12,13]. It is notable, there are a number of different definitions that can be considered. This gap is broadened by the absence of mechanistic biomarkers that allow identification and potential targeting of the pathophysiological alterations contributing to the development of sarcopenic obesity and frailty. The ability to quickly identify and monitor sarcopenic obesity and frailty in a cost- and time-efficient manner at the point-of-care can potentially be the first step in the rapid diagnosis beyond traditional body composition measures, which are often resource-intense.

This review outlines how clinical, radiographic, and biological data can be combined to optimize diagnostic and therapeutic strategies in older patients with sarcopenia and sarcopenic obesity. We further describe how the principles of geroscience and precision medicine can be leveraged to catalyze our ability to address the heterogeneity of aging [14,15] to improve physical function and reduce the risk of disability, enhance the quality of life, and reduce morbidity in older adults with sarcopenia, sarcopenic obesity, and frailty.

## 2. Biological Changes Contributing to the Development of Sarcopenic Obesity

### 2.1. Sarcopenic Obesity

The pathophysiology of sarcopenic obesity in older adults is complex, and there are many diverse contributing biological mechanisms (for a comprehensive review on the biological pathways to sarcopenic obesity, see Batsis and Villareal, 2018 [10]. Proposed model included here with permission of author and publisher (Figure 2)). Biological mechanisms contributing to the development of sarcopenic obesity can be categorized as follows: (a) body composition; (b) sex-specific hormones; (c) inflammatory pathways; and (d) myocellular pathways. Each of these are described briefly below.

### 2.2. Body Composition

Body composition changes occur with aging [16]. These include an increase in body fat until the seventh decade followed by a progressive decline, a reduction in height secondary to changes in vertebral architecture and a decline in muscle mass after the fourth decade such that most of the weight is gained as fat. This decrease in lean muscle mass is also associated with loss of strength. Intramuscular deposition of fat has been associated with decreased strength and function, and deposition rate increases with aging [17,18]. Factors contributing to these changes include age-related decreases in the components of energy expenditure, resting metabolic rate, physical activity, and the thermic effect of food [10].

### 2.3. Sex-Specific Hormones

Age-related changes in estrogen and testosterone are partly responsible for sex-specific changes in muscle and fat in older adults [19]. As estrogen is responsible for lipogenesis inhibition in muscle via fat oxidation [20,21], the decline in post-menopausal estrogen results in increases in total body weight, fat mass, and central adiposity with an associated reduction in fat-free mass in women [19]. In males, age-related declines in testosterone often have a negative impact on muscle mass and fat distribution via altered regulation of androgen receptors [22].

### 2.4. Inflammatory Pathways

Obesity is implicated in several inflammatory pathways [23]. These include upregulated mast cells, macrophages, and T lymphocytes, increasing the production of pro-inflammatory cytokines such as tumor necrosis factor and IL-6 [23,24]. These cytokines contribute to increased insulin resistance, which ultimately results in the loss of muscle mass and gain of fat mass [25].

### 2.5. Myocellular Pathways and Intramuscular Fat Deposition

Critically, all the changes above result in increased fatty infiltration of skeletal muscle [26]. Subsequently, intramyocellular lipids promote lipotoxicity and inflammation, induce differentiation of progenitor cells that express fatty tissue genes, and impair muscle regeneration and contractility by promoting fibrosis and impairing mitochondrial fatty acid oxidation [27]. The oxidative nature of intramuscular fat may also lead to worsening metabolic disease due to insulin resistance, leading to further fat deposition and inhibition of muscle formation [18,28]. Importantly, exercise in older adults slows or reverses many biological changes that occur with aging. Its effect includes upregulation of protein synthesis leading to increases in lean muscle mass, downregulating of inflammatory markers, and counteracting effects of intramyocellular lipids by reducing oxidative stress, improving muscle oxidative capacity, and improving fatty acid metabolism. The negative health impacts of sarcopenic obesity highlight the importance of understanding muscle quality versus size as intramuscular fat impacts strength outside of changes in muscle thickness [29]. Better understanding the contribution of intramuscular fat to the development of sarcopenic obesity may be at the crux of standardizing the definition of sarcopenic obesity, and ultimately, to clinical management and mitigation.

## 3. Biomechanics and Gait Performance

Intramuscular fat can potentially elicit a multi-scale musculoskeletal cascade that can impair gait, mobility, and independence (Figure 3). In summarizing those effects, we also emphasize needs and opportunities for innovation in our treatment and therapeutic interventions to delay, slow, and potentially reverse this cascade.

Intramuscular fat elicits numerous skeletal muscle adaptations that may ultimately contribute to decline in whole-muscle function and, thereby, gait and mobility. For example, intramyocellular fat content in older adults with obesity is associated with larger single fiber cross-sectional area and slower peak shortening velocity and specific power [30,31]. These impairments at the myofibril level subsequently contribute to reduced peak force from individual muscle fibers in persons with compared to those without obesity [30]. Intramuscular fat may also contribute to changes in whole-muscle properties and contractility as the infiltration of adipose tissue increases aggregate tissue stiffness, thereby limiting muscle fibers’ ability to bulge and shorten during a contraction with deleterious functional implications. Conversely, in vivo measurements suggest that higher levels of intramuscular fat are associated with lower shear modulus, an indirect proxy for tissue stiffness, at least in the distal leg muscles [32]. However, it is entirely plausible but poorly understood whether the extent of these multiscale effects differs between muscles, especially given the well-documented anatomical variation in architecture, morphology, and function. Moreover, these contradictory outcomes exemplify the need to improve our mechanistic insight into how intramuscular fat affects macroscale muscle mechanics and contractile performance.

Although it is intuitive for intramuscular fat to affect muscle properties and contractile performance, the transmission of muscle forces is simultaneously governed by the mechanics of series elastic tendon. Tomlinson et al. suggest that obesity does not associate with Achilles tendon mechanical properties among older adults [33]. However, given documented reductions in tendon stiffness due to age, deficient tendon adaptation to obesity may place individuals with sarcopenic obesity at a higher risk for musculotendinous injury. Tendon tissue integrity is influenced by the chronic low-grade inflammatory environment and accumulation of advanced glycation end products associated with sarcopenic obesity that can alter the mechanical behavior of tendons [34,35]. These physiological changes may alter the failure behavior of tendon fibers [35] resulting in altered force transmission and/or progressive connective tissue disorders among individuals with sarcopenic obesity.

Whole-body gait biomechanics such as walking speed and distance are routinely considered in the clinical assessment and objective measurement of physical capacity in people with sarcopenic obesity. However, gait biomechanics in those with obesity, per se, is considerably better understood than those in people with sarcopenic obesity. Those with sarcopenic obesity walk slower than age-matched individuals without sarcopenic obesity [36,37,38,39]. Older adults with lower appendicular muscle mass divided by total fat mass walk slower than people with higher total fat mass [36,37]. These results often extend to individuals diagnosed with sarcopenic obesity [38,39,40]. Given the relatively pervasive nature of this functional association, most clinical studies on individuals with sarcopenic obesity use a walking speed criterion of ≤0.8 m/s as a distinguishing characteristic [40,41]. Additionally, performance on the Timed Up and Go assessment was inferior in those individuals with sarcopenic obesity as compared to counterparts with normal appendicular skeletal muscle mass [42]. Similar detrimental effects were seen with poorer performance on the 6-min walking test, suggesting sub-maximal aerobic capacity [38]. While there are clinical and functional measures that support the diagnosis of sarcopenic obesity, it is unclear as to the impact of intramuscular fat on the musculotendinous architecture and whether advanced or point-of-care imaging can be used for supporting the diagnosis of sarcopenic obesity.

## 4. Imaging Modalities to Identify Intramuscular Fat

Though muscle function can be assessed clinically using conventional measures (e.g., grip strength, timed up and go, sit-to-stand tests), the differentiation between muscle quantity and quality is not easily assessed at the bedside. Non-invasive imaging biomarkers of muscle quantity and quality are available for clinical and research purposes. We will focus on imaging biomarkers of muscle fat infiltration, a significant predictor of muscle function independent of muscle volume (Table 1) [43,44]. Both direct and indirect measures of muscle fat are available, including quantitative fat fraction estimation by magnetic resonance imaging (MRI) and indirect measures using computed tomography (CT) and ultrasonography (US). While useful for measuring muscle quantity, dual energy X-ray absorptiometry (DEXA) is not able to measure fat fraction and therefore, intramuscular fat.

### 4.1. Magnetic Resonance Imaging

MRI is reliable and accurate for assessing muscle due to its inherent soft tissue contrast, as different tissues differ in their hydrogen density and relaxation time [45,46]. Muscle mass is often measured on T1-weighted images due to the large differences in signal intensity between fat (bright) and non-fat tissues (dark) [32]. Estimating fat infiltration using an intensity threshold on T1-weighted images assigns individual pixels to fat or muscle; however, this only accounts for macroscopic intramuscular fatty infiltration. Further accuracy can be determined by analyzing the differential signals from fat and water in a specific muscle. The fat fraction determined using this differential signal is the *proton density fat fraction* (PDFF). Dixon-based MRI can generate PDFF maps to assess entire muscles or even the entire body. MRI Spectroscopy is the gold standard for fat quantification. It is a localized technique that can further differentiate the slight frequency differences between different molecules to quantify the biochemical composition of a tissue (e.g., individual lipid moieties) [47]. Unlike Dixon-based PDFF, magnetic resonance spectroscopy can measure the intra-cellular lipid content by resolving slight differences between the intramyocellular and extramyocellular lipid peaks [48]. However, magnetic resonance spectroscopy cannot generate fat fraction maps and would not be reliable in situations where fat distribution may be heterogeneous. While MRI is highly accurate and avoids ionizing radiation, it is costly, resource intensive, requires length acquisition times and is susceptible to many artifacts [47]. From a scalability standpoint in clinical and research-based endeavors, its software is also not universally available.

### 4.2. Computed Tomography (CT)

CT creates a two-dimensional map of X-ray attenuation of the imaged body part with each pixel in the image given a value based on that attenuation specified in Hounsfield units (HU). Classically, measurements are obtained on a single slice in the abdomen (at L3) or the mid-thigh [49,50]. Many patients with chronic diseases obtain abdominopelvic CT scans for varied reasons. Thus, CT-based measures of muscle volume and quality may be assessed using an “opportunistic” assessment of muscle quality at no additional imaging cost. Segmenting muscles also allows the determination of the muscle cross-sectional area and computation of the skeletal muscle index [50]. The anatomic level of assessing muscle is agreed upon at L3. Yet, there is considerable variation in CT acquisition techniques [51].

Specifically, the presence and phase of intravenous contrast (arterial versus portal venous versus delayed) can result in significant changes in attenuation values. Both standard/diagnostic or low-dose CT may also lead to different values. Finally, CT is considered less sensitive than MRI for detecting fat and cannot resolve fat fractions of less than 5% [52,53]. This lack of granularity is a barrier in intervention-based research where small changes may be relevant [1,54].

### 4.3. Ultrasonography

Ultrasonography (US) is based on high frequency sound waves penetrating tissues and then analyzing the reflections to produce an image. Such equipment is portable, relatively inexpensive, and is not associated with ionizing radiation making US an attractive choice for the point-of-care assessment of muscle quantity and quality. For instance, US is being routinely used to guide bedside procedures [55,56,57,58] and in the assessment of volume status (e.g., pulmonary edema, ascites) [58,59], skin and soft tissue infections [60], and basic cardiac function by internists [59], leading to support statements for use by both the American College of Physicians and Society of Hospital Medicine for patient assessments and procedures [61].

Muscle quality is based largely on the echointensity (EI) of B-mode ultrasound images [44,62]. Adipose tissue and intramuscular fat are both hyperechoic (bright) on ultrasound, whereas skeletal muscle, particularly the contractile components, appears hypoechoic (dark) interspersed with bright fascial and tendinous elements. EI by ultrasound correlates with muscle quality, but it is not a direct measure of fat infiltration [54]. Elevated EI is not only a result of intramuscular fat, but also fibrosis or edema, and likely increases in all muscles with age as a result of medical co-morbidity (e.g., heart failure, osteoarthritis) [44]. As such, alternative explanations for elevated EI need to be excluded before assessing muscle quality in the assessment of sarcopenia and frailty. While EI is easily measured, many confounding factors need to be accounted for: numerous machine-dependent settings affect EI, and increased thickness of overlying adipose tissue results in decreased EI [17,44,63]. Since we are concerned with the interplay between muscle quality and obesity, the effect of subcutaneous adipose tissue thickness of EI cannot be ignored. Additionally, an oft-cited drawback of US is operator dependence [44]. While assessment of muscle volume is relatively user-independent [50], evaluation of EI is highly dependent on operator factors. To maintain consistency, a single trained operator should be used for EI measurements, especially in research [44]. Despite progress in this area, following changes in a single subject will be more reliable than drawing conclusions in a population that includes large differences in the amount of overlying adipose tissue. Identifying intramuscular fat at the bedside is important for identifying individuals at risk for sarcopenic obesity and may benefit from targeted interventions such as exercise programs and addressing specific aging pathways and biomarkers.

## 5. Integrating Biological Measures to Identify and Target Mechanistic Pathways

The pathophysiology of geriatric syndromes, such as physical frailty and sarcopenia, cannot be traced to a single defect, but instead results from the integration of dysfunction across multiple organ systems [64]. Therefore, understanding molecular pathologies of age-related diseases and providing interventions targeted at the fundamental mechanisms may hold promise in preventing and delaying the onset and progression of sarcopenia and other age-related diseases [7,65,66,67]. The cutting-edge multi-omics methodologies including genomics, transcriptomics, proteomics and metabolomics/exposomics, together with bioinformatics techniques such as regression modeling and artificial intelligence, have provided novel tools to understand the interactions between host metabolism and lifetime exposures (Table 2) [64]. These exposures include dietary intake, which may play an important role in molecular mechanisms underlying sarcopenia and frailty. The multi-omics approaches could generate two major types of findings: (1) biomarker(s) in gene, protein and/or metabolite level, which have potential to be used for early detection and diagnosis, prediction of sarcopenia related conditions; and (2) perturbations in biological pathways pointing to physiological mechanisms underlying syndromes, and thereby informing interventions targeted at fundamental disease processes. In recent years, there have been increased efforts to harness the principles of geroscience, heterogeneity, and biological models to further understand the aging process [68]. Biological and metabolic pathways have mostly been proposed in mice models, providing frameworks for exploring specific biomarkers and targeted pharmacological interventions in aging adults [69]. Examples include targeting frailty phenotypes in mice with Janus kinase inhibitors and potentially predictive molecular markers of aging such as p16IN^K4a^ and Interleukin-6 [70]. While p16IN^K4a^ in mice accumulates throughout the lifespan, there is no apparent effect on a mortality endpoint. Therefore, further exploration of biomarkers as potential predictors of aging and endpoints are needed in mice and human models. A limitation to human exploration includes FDA-approved tests measuring various biomarkers [70].

Muti-omics approaches have emerged in recent years as means of capturing and underling the complex pathophysiology of geriatric syndromes [71,72,73]. Genomics and proteomics research to date have revealed that Caveolin protein 1 mutations and functional decline of carboxylate proteins may contribute to the pathophysiology of sarcopenia [74,75,76]. Additionally, among available analytical tools, metabolomics can be used to identify and quantify the repertoire of small molecules from a wide variety of biological samples (e.g., cells, tissues, and biological fluids). The collection of metabolites, termed the metabolome, may define the final output of genome-environment interactions and is highly dynamic and individualized [77]. Metabolomics has been used to demonstrate the concept of a “sarcopenic phenotype” [78] and to identify plasma traumatic acid as a potential biomarker in sarcopenia [72]. Pathways associated with sarcopenic phenotype include biosynthesis of amino acids, arginine and proline metabolism, the biosynthesis of alkaloids derived from ornithine, linoleic acid metabolism, and the biosynthesis of unsaturated fatty acid [78]. Several studies have assessed the predictive validity of untargeted metabolomics in the diagnosis of sarcopenia and frailty. Pujos-Guillot et al. used untargeted metabolomics of serum obtained from the European NU-Age study to better characterize the complexity of the pre-frailty phenotype and to identify an array of biomarkers that may predict frailty trajectory [79]. For prefrail subjects who improved their status over time, modeling of the metabolomics data revealed four metabolites, for each sex, that resulted in strong predictability for men [AUC, 0.93 (95% CI = 0.87–1)] and women [AUC, 0.94 (95% CI = 0.87–1)]. Three sex specific predictive markers of pre-frailty were identified by modelling metabolic profiles of non-frail participants who became pre-frail overtime. This also resulted in very good AUCs for men (AUC = 0.82; 95% CI = 0.72–0.93) and for women (AUC = 0.92; 95% CI = 0.86–0.99). While these investigations were conducted with small numbers of patients for each of the phenotypic groups, they demonstrate the promise for metabolomics to deliver early and predictive markers of frailty, and markers to monitor the progression or resolution of disease.
nutrients-14-01384-t002_Table 2Table 2Multi-OMICS in sarcopenia and frailty.StudyModelCategorySyndromeSummary of Study Findings Regarding Biomarkers, Metabolic Pathways, and Gene Associated with AgingLin et al. [76]HumanGenomicsSarcopeniaThe A allele of the CAV1G14713A Caveolin protein 1 (CAV1) may be a predictor for higher likelihood of developing sarcopenia and severe sarcopenia in a Taiwanese older adult population.Dos Santos et al. [74]HumanProteomicsSarcopeniaThe functional decline in 17 carboxylate proteins involved in cellular transport, energy metabolism and muscle contraction may be associated with a sarcopenic phenotype.Tsai et al. [72]HumanMetabolomicsSarcopeniaPlasma traumatic acid has been identified as potential biomarker for sarcopenia.Opazo et al. [78]HumanMetabolomicsSarcopeniaPathways of biosynthesis of amino acids and alkaloids derived from ornithine, arginine and proline metabolism, linoleic acid metabolism, and the biosynthesis of unsaturated fatty acids are associated with a “sarcopenic phenotype.”Pujols et al. [79]HumanMetabolomicsPre-frailtyFour potential markers for each sex that discriminate between sub-phenotypes of pre-frailty.Men: glutamine, glycine-phenylalanine, dimethyloxazole, mannoseFemale: threonine, fructose, mannose, N-(2-hydroxylpropyl)-valineBurd et al. [70]MiceProteomicsFrailtyMice models suggest that molecular markers associated with aging, such as p16IN and IL6, are potential targets for pharmacological interventions using Janus kinase inhibitors.

Incorporating multi-omics platforms with clinical and radiographic findings using artificial intelligence can further enhance the mechanistic processes involved in the heterogeneity of aging, a key tenant of geroscience, and allowing for targeting patient-specific interventions (e.g., precision medicine) [14,66].

## 6. Nutritional Interventions

Current status quo includes multicomponent interventions that focus on diet and exercise. This section will focus on nutritional interventions, consisting of caloric restriction and other supplements that could potentially reduce intramuscular fat deposition and potentiate the synergistic improvement in muscle mass and strength.

Caloric restriction is the cornerstone of therapy. Specifically, it has been shown that dietitian-based interventions successfully lead to fat mass loss—both subcutaneous and visceral. Villareal and Batsis’ interventions reduced caloric intake in older participants with obesity by at least 500–750 kcal/day, with a minimum of 1200 kcal. [80,81] This is of critical importance as a reduction in excess of these amounts may disproportionately lead to more loss of muscle mass and strength than a corresponding loss of fat. Heymsfield has advocated that a loss of 1.0 kg in weight amounts to 75% fat mass, and 25% muscle mass [82]. As outlined above, additional muscle loss may promote and exacerbate physical functional impairments.

The type of diet remains elusive. The above interventions, and those of others, often focus on macronutrient rather than micronutrient composition. Evidence suggests that in older adults, the two diets that may have the best impact on body composition include the Dietary Approaches to Stopping Hypertension (DASH) and the Mediterranean Diet. Both have a favorable impact on metabolic variables, including reducing many of the abovementioned components presented in Figure 2. Yet, in epidemiological studies, Soltani did not demonstrate any impact of the DASH diet on risk of sarcopenia [83]. Similarly, this was not observed in a study evaluating the Mediterranean diet [84]. However, both of these studies were cross-sectional, so it would be difficult to ascertain any type of causative effect. Alternatively, in a small study of maintained 1800 kcal/day for 12-weeks under controlled conditions, a DASH diet led to reductions in body fat percentage, fat mass, and maintained grip strength, gait and balance, all suggesting the potential for DASH to preserve muscle strength while reducing fat mass in older adults with obesity [85]. In a subsequent analysis, these authors also demonstrated favorable alterations as a result of this diet on key inflammatory mediators that have implications in sarcopenic obesity.

Reduction in intramuscular fat is believed to be the harbinger of much of what is described above. Medications, including PPAR-g agonists, while decreasing abdominal visceral adipose tissue, have been shown to reduce intramuscular fat within the thigh based on CT-findings. While exercise may preferentially reduce intramyocellular adipose tissue, in conjunction with calorie restriction, such fat deposition. [86] In this intervention, there was a reduction in calorie intake by 16% in the first three months, followed by 20% in the remaining 9 months.

Other potential nutrients that have been recommended by consensus, rather than by robust clinical trial evidence, include omega-3 or omega-6 supplementation (including fish-oil), vitamin D, and dietary strategies that affect insulin resistance. Unfortunately, dietary protocols, duration of calorie restriction, and type of micro/macronutrient composition in longitudinal studies are critically required to develop effective interventions to reduce or reverse myosteatosis that can clinically signify improvement indicators.

## 7. Gaps in the Science

Translation of research findings into current clinical practice is impeded by an inability to quickly identify and monitor sarcopenia in a cost- and time-efficient manner at the point-of-care. In our assessment of the available literature, there is a dearth of studies that quantify the biomechanical features of gait quality, including step length, width, and frequency, ground reaction forces, joint mechanics and energetics, and patterns of muscular recruitment in people with sarcopenic obesity. Historically, these measures would be limited to laboratory environments with sophisticated and expensive equipment. However, the evolution and widespread adoption of wearable sensors continues to break down barriers to clinical translation [87,88]—increasing the feasibility of these measures in the clinic. Ultimately, studies into the mechanistic causal associations between muscle-level changes and gait performance outcomes are warranted into order to identify potentially modifiable factors and therapeutic interventions. This critical gap is broadened by the absence of mechanistic biomarkers to bridge point-of-care diagnostics to the prescription of effective, potentially precision medicine-based interventions. This will allow individualistic targeting of the specific intervention to the specific person using baseline characteristics that include biological and functional measures in advanced machine-learning based models. Such analytical approaches permit researchers (and ultimately clinicians) to incorporate a multitude of covariates into their decision support systems.

There is an urgent need for high-quality evidence to support specific and tailored strategies to reduce the risk of disability, in contrast to a one-size-fits-all approach. Creating optimal diagnostic and therapeutic strategies will ensure adequate allocation of resources, allowing providers to identify at-risk older adults at the bedside by incorporating geroscience principles to address the heterogeneity of aging [14]. Combining data sources will allow for targeting interventions to the right person (e.g., precision medicine) with the goals of maximizing resource allocation and minimizing unnecessary interventions for at-risk older adults. This approach has the potential to improve physical function using personalized, evidence-based interventions to enhance quality of life, reduce morbidity, and reduce the risk of disability.

## 8. Conclusions

Quantitative biomechanics and imaging reliably and efficiently assess lean muscle mass and function. We hypothesize that, when computationally coupled with multi-omics platforms, this approach has the potential to advance translational geroscience by offering greater insights into the heterogeneity of aging processes. Specifically, this integration will allow us to move past the challenges of defining sarcopenia and sarcopenic obesity to better identify and develop interventions to prevent incident frailty. Gaining an understanding of biological mechanisms and targeting them has a strong potential to lay the foundation for the development of precision medicine interventions that will improve the lives of many older adults.

## Figures and Tables

**Figure 1 nutrients-14-01384-f001:**
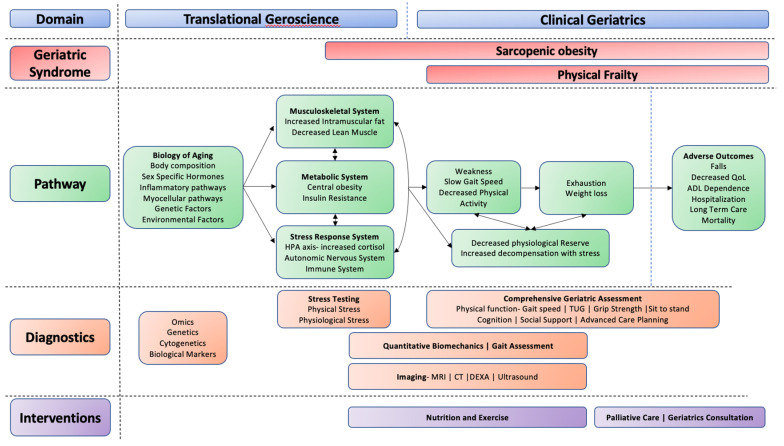
Geroscience conceptual model. Proposed pathways of exploration using translational geroscience (e.g., biological pathways) and clinical geriatrics (e.g., functional assessments and imaging) to identify sarcopenic obesity and physical frailty and develop targeted interventions for mitigation.

**Figure 2 nutrients-14-01384-f002:**
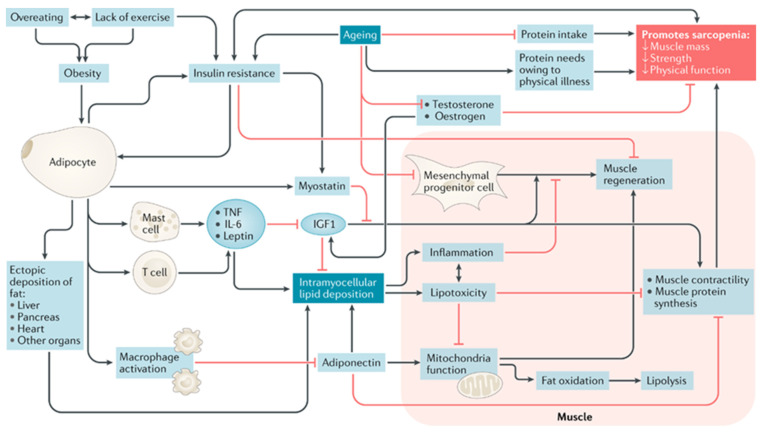
A proposed model of mechanisms leading to sarcopenic obesity. The proposed interplay between adipose and muscle tissue, which is believed to contribute to the development of sarcopenic obesity, is shown. The black lines are stimulatory, while red lines with flat ends indicate inhibition. IGF1, insulin-like growth factor 1; TNF, tumor necrosis factor [10].

**Figure 3 nutrients-14-01384-f003:**
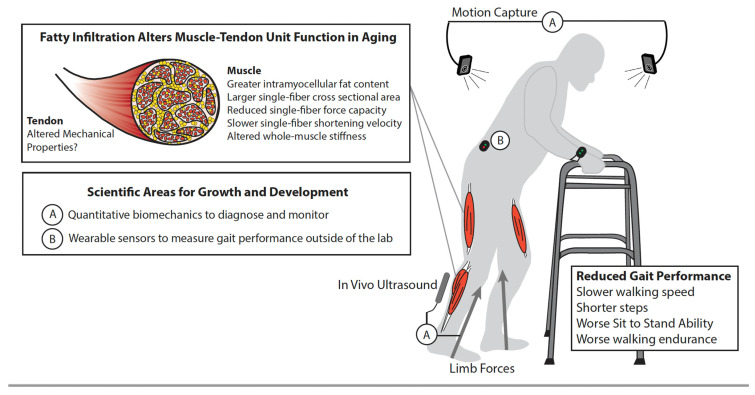
Multi-scale biomechanics and gait performance. Schematic illustrates the musculoskeletal cascade from the accumulation of intramuscular fat to reduced gait performance among older adults. This is also simultaneously emphasized, and discussed in the narrative: (**A**) the need for quantitative biomechanics to objectively diagnose and monitor the salient features of gait quality, including step length, width, and frequency, ground reaction forces, joint mechanics and energetics, and patterns of muscular recruitment in people with sarcopenic obesity, and (**B**) the evolution and widespread adoption of wearable sensors that continue to break down barriers to clinical translation.

**Table 1 nutrients-14-01384-t001:** Muscle fat infiltration estimation by modality.

Modality	Method	Pros	Cons
MR	Threshold Intensity: sets intensity threshold on T1-weighted images to assign pixels to muscle or fat categories	Requires no specialized imaging sequences Obtained routinely on all MRI examinations	Costly Only measures macroscopic fat (Intermuscular adipose tissue)
	Chemical Shift Methods: Based on the difference between different molecular resonance frequencies (i.e., fat and water)	Reliable and accurate Used as comparison standard	Costly Long acquisition time Requires specialized sequences that are not universally available
	Dixon: generates images based on the constructive and destructive interference between water and fat	Can generate FF map Accurate across vendors, imaging centers, and field strengths Reproducible	Susceptible to artifacts (respiratory/motion, magnetic field inhomogeneity due to metallic implants) Need for multipoint Dixon and post-processing to account for artifacts
	MR Spectroscopy: calculated based on the area under the lipid peak produced, for a single voxel	Identify individual lipid moieties Can distinguish intra-cellular and extra-cellular fat	Large voxel size needed Cannot be used to generate fat fraction maps Not reliable in heterogenous fat distribution leading to low short-term reproducibility
CT	2D maps of pixels of different X-ray attenuation (HU). FF = inversely proportional to muscle attenuation in region of interest (darker muscle = more fat).	Can generate density map that correlates with fat fraction Best for intermuscular adipose tissue quantification Abdominopelvic CTs often acquired for other reasons	Less sensitive than MRI Unable to resolve FF < 5% Radiation Attenuation mapping needs to be established using MRI (PDFF) Influenced by variability in acquisition (i.e., presence of IV contrast and phase of contrast, tube voltage and current which may be adjusted for patients with higher BMI)
US	Based on muscle echointensity (higher echointensity = increased fat/decreased quality)	Can be portable/bedside No ionizing radiation Correlates with MR	EI not true estimate of fat fraction (more of muscle “quality” indicator) and is influenced by many other factors Influenced by depth/transducer frequency Overestimates quality in patients with more overlying tissue without correction Highly operator dependent, no standardized acquisition method
DEXA	Cannot determine fat fraction or muscle quality—only percent of lean muscle mass		

## Data Availability

Not applicable.

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
