# Peer review of "Multimodal Diagnostic Approaches to Advance Precision Medicine in Sarcopenia and Frailty"

_nutrients, 2022, doi:10.3390/nu14071384_

Round 1

Reviewer 1 Report

Title: Multimodal Diagnostic Approaches to Advance Precision Medicine in Sarcopenia and Frailty
David H. Lynch BMBS1 , Hillary B. Spangler MD2 , Jason R. Franz 3 , Rebecca L. Krupenevich 3 , Hoon Kim 3 , Daniel 4 Nissman MD MPH MSEE4 , Janet Zhang MD4 , Yuan-Yuan Li5 , Susan Sumner5 , John A. Batsis and MD1,5
3/9/2022
First let me express appreciation to the Editor for the opportunity to review the interesting manuscript which perhaps could be more descriptively titled: Multimodal Diagnostic Approaches to Sarcopenia and Fraility

Or even more conventionally: A Review of Sarcopneia and Fraility

I also think starting off with a Table of Contents would be helpful, as often done in topic reviews in the medical literature:

1.    A basic model for geroscience
   A reference to perhaps include - Sierra F, Caspi A, Fortinsky RH, Haynes L, Lithgow GJ, Moffitt TE, Olshansky SJ, Perry D, Verdin E, Kuchel GA. Moving geroscience from the bench to clinical care and health policy. J Am Geriatr Soc. 2021 Sep;69(9):2455-2463. doi: 10.1111/jgs.17301. Epub 2021 Jun 19. PMID: 34145908.

2.    Sarcopenic obesity
a.    This seems to have been just published and could be included;
Bartoli F, Debant M, Chuntharpursat-Bon E, Evans EL, Musialowski KE, Parsonage G, Morley LC, Futers TS, Sukumar P, Bowen TS, Kearney MT, Lichtenstein L, Roberts LD, Beech DJ. Endothelial Piezo1 sustains muscle capillary density and contributes to physical activity. J Clin Invest. 2022 Mar 1;132(5):e141775. doi: 10.1172/JCI141775. PMID: 35025768.

3.    BIOMECHANICS AND GAIT PERFORMANCE 
4.    Imaging Modalities
a.    This section is much too detailed and technical at least for this reader.
5.    Multi-omic studies 
6.    Nutritional Interventions
7.    Current clinical practice recommendations
8.    A view toward precision medicine in geriatrics Ie sarcopenia and frailty will be a part of the clinical approach. [ Or maybe only an injection of Piezo1 will suffice, sic]

On a literary note, the language could be simplified to advantage, eg.

Quantitative biomechanics and imaging can be used to assess lean muscle mass and 422 muscle function reliably and efficiently.

Could be replaced with :Quantitative biomechanics and imaging reliably and efficiently assess lean muscle mass and function.    

On another note, the authors do not consider anthropometric assessment for sarcopenia and grip strength, which at least have the cost advantage. (1-4)

1.    Cho HW, Chung W, Moon S, Ryu OH, Kim MK, Kang JG.   Effect of Sarcopenia and Body Shape on Cardiovascular Disease According to Obesity Phenotypes.   Diabetes Metab J. 2020 Jan;44:e38.   

2.    Tay L, Ding YY, Leung BP, Ismail NH, Yeo A, Yew S, Tay KS, Tan CH, Chong MS. (2015)  Sex-specific differences in risk factors for sarcopenia amongst community-dwelling older adults.201Age (Dordr).5 Dec;37(6):121. doi: 10.1007/s11357-015-9860-3. 

3.    Krakauer NY, Krakauer JC. Association of Body Shape Index (ABSI) with Hand Grip Strength. Int J Environ Res Public Health. 2020 Sep 17;17(18):E6797. doi: 10.3390/ijerph17186797. PMID: 32957738.

4.     Krakauer, N.Y.; Krakauer, J.C. Association of X-ray Absorptiometry Body Composition Measurements with Basic Anthropometrics and Mortality Hazard. Int. J. Environ. Res. Public Health 2021, 18, 7927. https://doi.org/10.3390/ijerph18157927

Some line by line comments follow: 

until the seventh decade followed by a progressive decline, a reduction in height because 91 of vertebral compression a  91,92 – sounds false -  only if due to osteoporosis – most height loss is due to loss of intervertebral disc space, spinal curvatures such as dorsal kyphosis  etc Because estrogen is responsible for lipo- 105 genesis inhibition in muscle via fat oxidation,20 the decline in post-menopausal estrogen 106 results in increases in total body weight, fat mass, and central adiposity with an associated 10 should use a bit more definitive reference that #20!
Found this:

Tara M. D'Eon, Sandra C. Souza, Mark Aronovitz, Martin S. Obin, Susan K. Fried, Andrew S. Greenberg, Estrogen Regulation of Adiposity and Fuel Partitioning: EVIDENCE OF GENOMIC AND NON-GENOMIC REGULATION OF LIPOGENIC AND OXIDATIVE PATHWAYS*,
Journal of Biological Chemistry, Volume 280, Issue 43, 2005, Pages 35983-35991,

These impairments at the myofibril level subsequently contribute to reduced peak force 150 
from individual muscle fibers in persons with compared to those without obesity.30 Intra- 151

beyond line 151: The entire section 146-200 needs to be simpler and MUCH more concise 

While useful for measuring muscle quantity, dual energy 211 X-ray absorptiometry (DEXA) is not able to measure fat fraction. 212---DXA measures fat and lean (nonfat)  mass of limb, but not % muscle fat

Table 1. Muscle Fat Infiltration Estimation by Modality DEXA Cannot determine fat fraction or muscle quality—only muscle mass----NO – DXA does not determine “muscle mass” -  determines non fat – designated lean mass which includes, blood, vessels etc it has been empirically validated to correlate very well with muscle mass – carcass studies can be referenced.
The entire radiology section 202-289    is way more technical than this reviewer can assess and seems inappropriate for “Nutrients”.

Author Response

March 18, 2022

To the Editor,

Thank you for reviewing our manuscript, Multimodal Diagnostic Approaches to Advance Precision Medicine in Sarcopenia and Frailty. We appreciate your thoughtful comments. We’ve addressed the comments below with corresponding changes via Track Changes in the manuscript document:

First let me express appreciation to the Editor for the opportunity to review the interesting manuscript which perhaps could be more descriptively titled: Multimodal Diagnostic Approaches to Sarcopenia and Frailty Or even more conventionally: A Review of Sarcopenia and Frailty

Thank you for your suggestion. Because of the importance of precision medicine in addressing the current (and many) gaps in geroscience and diagnosis of sarcopenia and frailty, efforts to mitigate sarcopenia and frailty are the focus of the piece. We can of course change the title if editors and reviewers feel it would better reflect our work. However, we feel the current title will lead to increased engagement from researchers in fields of precision, medicine, and geroscience since our aim was true integration of these topics rather than duplicating efforts that have been previously published.

I also think starting off with a Table of Contents would be helpful, as often done in topic reviews in the medical literature: 

Thank you for this suggestion. We have included a table of contents to assist the reader with navigation of the article based and changed the headings based on your suggestions below (in bold).

  1.    A basic model for geroscience A reference to perhaps include - Sierra F, Caspi A, Fortinsky RH, Haynes L, Lithgow GJ, Moffitt TE, Olshansky SJ, Perry D, Verdin E, Kuchel GA. Moving geroscience from the bench to clinical care and health policy. J Am Geriatr Soc. 2021 Sep;69(9):2455-2463. doi: 10.1111/jgs.17301. Epub 2021 Jun 19. PMID: 34145908.

This is a nice addition to the article to highlight the importance and gaps within geroscience. Article reference added into manuscript.

  1.    Sarcopenic obesity
    a.    This seems to have been just published and could be included;

    Bartoli F, Debant M, Chuntharpursat-Bon E, Evans EL, Musialowski KE, Parsonage G, Morley LC, Futers TS, Sukumar P, Bowen TS, Kearney MT, Lichtenstein L, Roberts LD, Beech DJ. Endothelial Piezo1 sustains muscle capillary density and contributes to physical activity. J Clin Invest. 2022 Mar 1;132(5):e141775. doi: 10.1172/JCI141775. PMID: 35025768.

Thank you for the recommendation. This fits nicely into the geroscience section as noted below, as it highlights an additional biomarker of aging worth (line 323).

  1.    BIOMECHANICS AND GAIT PERFORMANCE
    4.    Imaging Modalities
    a.    This section is much too detailed and technical at least for this reader.

Thank you for this feedback. We do think some of the granular details are important to present. We detailed biomechanical analysis in older adults with functional impairment have the potential to improve our identification of intramuscular fat and thus further our understanding of its role in the development in sarcopenic obesity and frailty. Lack of imaging protocols is one of the major gaps in terms diagnosis (via imaging) of sarcopenia/sarcopenic obesity and we believe this section can meaningfully add to the literature. By including a more detailed view in the section, we hope to introduce the reader to imaging-specific concepts that can be incorporated in exploration of bed-side accessible imaging and downstream interventions. If the reviewer feels there are specific details that are outside the scope of our overall intent, we would be most happy to make those changes.

  1.   Multi-omic studies 
    6.    Nutritional Interventions
    7.    Current clinical practice recommendations
    8.    A view toward precision medicine in geriatrics Ie sarcopenia and frailty will be a part of the clinical approach. [ Or maybe only an injection of Piezo1 will suffice, sic]

As above, added to line 323.

On a literary note, the language could be simplified to advantage, eg.

Appreciate clarification as to which part of the manuscript this is referring to.

Quantitative biomechanics and imaging can be used to assess lean muscle mass and 422 muscle function reliably and efficiently.  Could be replaced with :Quantitative biomechanics and imaging reliably and efficiently assess lean muscle mass and function.    

Thank you for this suggestion. We have incorporated this change in our manuscript.

On another note, the authors do not consider anthropometric assessment for sarcopenia and grip strength, which at least have the cost advantage. (1-4)

Grip strength as component of diagnosis mentioned in lines 68, 212; DEXA mentioned line 70. We are including Donini LM, Busetto L, Bischoff SC, Cederholm T, Ballesteros-Pomar MD, Batsis JA, Bauer JM, Boirie Y, Cruz-Jentoft AJ, Dicker D, Frara S, Frühbeck G, Genton L, Gepner Y, Giustina A, Gonzalez MC, Han HS, Heymsfield SB, Higashiguchi T, Laviano A, Lenzi A, Nyulasi I, Parrinello E, Poggiogalle E, Prado CM, Salvador J, Rolland Y, Santini F, Serlie MJ, Shi H, Sieber CC, Siervo M, Vettor R, Villareal DT, Volkert D, Yu J, Zamboni M, Barazzoni R. Definition and Diagnostic Criteria for Sarcopenic Obesity: ESPEN and EASO Consensus Statement. Obes Facts. 2022 Feb 23:1-15. doi: 10.1159/000521241. Epub ahead of print. PMID: 35196654 as a supportive reference for sarcopenia evaluation and diagnosis.

  1.    Cho HW, Chung W, Moon S, Ryu OH, Kim MK, Kang JG.   Effect of Sarcopenia and Body Shape on Cardiovascular Disease According to Obesity Phenotypes.   Diabetes Metab J. 2020 Jan;44:e38.   
  2.    Tay L, Ding YY, Leung BP, Ismail NH, Yeo A, Yew S, Tay KS, Tan CH, Chong MS. (2015)  Sex-specific differences in risk factors for sarcopenia amongst community-dwelling older adults.201Age (Dordr).5 Dec;37(6):121. doi: 10.1007/s11357-015-9860-3. 
  3.    Krakauer NY, Krakauer JC. Association of Body Shape Index (ABSI) with Hand Grip Strength. Int J Environ Res Public Health. 2020 Sep 17;17(18):E6797. doi: 10.3390/ijerph17186797. PMID: 32957738.
  4.     Krakauer, N.Y.; Krakauer, J.C. Association of X-ray Absorptiometry Body Composition Measurements with Basic Anthropometrics and Mortality Hazard. Int. J. Environ. Res. Public Health 2021, 18, 7927. https://doi.org/10.3390/ijerph18157927

Some line by line comments follow: 

until the seventh decade followed by a progressive decline, a reduction in height because 91 of vertebral compression a  91,92 – sounds false -  only if due to osteoporosis – most height loss is due to loss of intervertebral disc space, spinal curvatures such as dorsal kyphosis  etc

Changes made within manuscript to better explain the changes in the vertebral architecture as a whole, line 100-101.

Because estrogen is responsible for lipo- 105 genesis inhibition in muscle via fat oxidation,20 the decline in post-menopausal estrogen 106 results in increases in total body weight, fat mass, and central adiposity with an associated 10 should use a bit more definitive reference that #20!
Found this:

Tara M. D'Eon, Sandra C. Souza, Mark Aronovitz, Martin S. Obin, Susan K. Fried, Andrew S. Greenberg, Estrogen Regulation of Adiposity and Fuel Partitioning: EVIDENCE OF GENOMIC AND NON-GENOMIC REGULATION OF LIPOGENIC AND OXIDATIVE PATHWAYS*,
Journal of Biological Chemistry, Volume 280, Issue 43, 2005, Pages 35983-35991,

We greatly appreciate this suggestion – this reference explains the biochemical pathways related to estrogen and supports our statements about estrogen and the development of sarcopenic obesity. We have incorporated this as a citation within the manuscript, line 116.

These impairments at the myofibril level subsequently contribute to reduced peak force 150 
from individual muscle fibers in persons with compared to those without obesity.30 Intra- 151
 beyond line 151: The entire section 146-200 needs to be simpler and MUCH more concise 

Thank you for this feedback. We have reviewed this section and made several edits. We condensed explanations at the muscular level and associated clinical correlates, lines 156-209. We accept that this section remains lengthy and detailed however, we feel this is another major gap in the understanding of the relationship between sarcopenic obesity and muscle function. Noting these gaps are important as they further highlight the need to understand underlying muscle architecture.

While useful for measuring muscle quantity, dual energy 211 X-ray absorptiometry (DEXA) is not able to measure fat fraction. 212---DXA measures fat and lean (nonfat)  mass of limb, but not % muscle fat

Added “is not able to measure fat fraction and therefore, intramuscular fat.” to clarify explanation, lines 219-221.

Table 1. Muscle Fat Infiltration Estimation by Modality DEXA Cannot determine fat fraction or muscle quality—only muscle mass----NO – DXA does not determine “muscle mass” -  determines non fat – designated lean mass which includes, blood, vessels etc it has been empirically validated to correlate very well with muscle mass – carcass studies can be referenced.

Clarified within chart as below and adding Connaughton et al article for support.

Cannot determine fat fraction or muscle quality—only percent of lean muscle mass

The entire radiology section 202-289    is way more technical than this reviewer can assess and seems inappropriate for “Nutrients”.

Thank you again for your review of this section. We have made some changes to this section that simplify some of the technical components. Continuing the response from 4a above, we feel that it is important for readers to understand the gaps in identification of and intervention for sarcopenia/sarcopenic obesity. Moving forward, we believe that the best clinical approaches to sarcopenic obesity will involve a better understanding of imaging and how this also can lead to more patient-friendly imaging modalities. We trust that our suggested changes are satisfactory to the reviewer and ultimately to the reader to gain a larger perspective of the novelty of our approach

Reviewer 2 Report

In this review the authors give an extensive description of updated parameters (clinical, radiographic  and biological datas) to evaluate Sarcopenia and Frailty  in older adults. The manuscript presents a good overview on the topic, refering to appropriate studies as well as  references, and it will be  really interesting and usuful for clinicians

Author Response

Thank you very much for your review and kind words.